# Access to Healthcare during COVID-19

**DOI:** 10.3390/ijerph18062980

**Published:** 2021-03-14

**Authors:** Alicia Núñez, S. D. Sreeganga, Arkalgud Ramaprasad

**Affiliations:** 1Department of Management Control and Information Systems, School of Economics and Business, Universidad de Chile, Santiago 8330015, Chile; 2Ramaiah Public Policy Center, Bengaluru 560054, India; sreeganga.sd@rppc.ac.in (S.D.S.); arkalgud.ramaprasad@rppc.ac.in (A.R.); 3Information and Decision Sciences Department, University of Illinois at Chicago, Chicago, IL 60607, USA

**Keywords:** health equity, healthcare access, ontology, COVID-19

## Abstract

Ensuring access to healthcare is critical to prevent illnesses and deaths from COVID-19 and non-COVID-19 cases in health systems that have deteriorated during the pandemic. This study aims to map the existing literature on healthcare access after the appearance of COVID-19 using an ontological framework. This will help us to formalize, standardize, visualize and assess the barriers to and drivers of access to healthcare, and how to continue working towards a more accessible health system. A total of 131 articles are included and considered for mapping in the framework. The results were also compared to the World Health Organization guidelines on maintaining essential health services to determine the overlapping and nonoverlapping areas. We showed the benefits of using ontology to promote a systematic approach to address healthcare problems of access during COVID-19 or other pandemics and set public policies. This systematic approach will provide feedback to study the existing guidelines to make them more effective, learn about the existing gaps in research, and the relationship between the two of them. These results set the foundation for the discussion of future public health policies and research in relevant areas where we might pay attention.

## 1. Introduction

The COVID-19 pandemic continues to be a major global public health threat, challenging the provision of healthcare services and their accessibility. It has even affected those countries with high availability of healthcare facilities, cutting edge technologies, and a reasonable number of healthcare professionals. Therefore, regardless of the country or continent, all have had to adapt their systems to prompt access and find the best way to respond to this virus.

Healthcare access refers to the ease with which individuals can obtain needed healthcare. It is generally defined as the opportunity to use appropriate services in proportion to healthcare needs [1,2]. If services are available, then an opportunity exists to obtain medical care; however, it is also limited by other barriers such as financial, organizational, social, cultural issues, etc. [3]. In this sense, the level of access influences the use of medical services, and therefore the health status of the population. Access was a problem prior to the pandemic. As of today, there is preliminary evidence of racial and socio-economic disparities in the population affected by COVID-19 [4,5] due to the reduction in access to and utilization of healthcare services. As a result, inadequate or inaccessible access to healthcare services has exacerbated the existing social disadvantages, stressing the system even more.

Many resources and staff are being diverted from their normal activities to test and provide treatment for COVID-19 cases. Supplies are limited and people fear accessing healthcare providers [6]. Nowadays, the population is also starting to fear the effects from the COVID-19 vaccine. Therefore, it is essential to ensure access to medical care to prevent illnesses and deaths from COVID-19 and non-COVID-19 cases in already weak health systems. The reinforcement of strategies and the establishment of proactive measures to ensure that access to healthcare is not disrupted is important to mitigate the effects and spread of COVID-19 [7].

Reduced access to care, surgeries, and other hospital services, combined with fear of exposure to the virus, have led to a significant drop in access. Thus, many diseases that develop symptoms have been treated by using telemedicine. Telemedicine has surged as a feasible tool to maintain patient care and reduce the risk of COVID-19 exposure to patients, healthcare workers, and the public [8,9]. There is evidence of patients who have been managed by using telemedicine and expressed satisfaction with the services received, demonstrating that telemedicine helped assessing, diagnosing, triaging, and treating patients with COVID-19 while avoiding a visit to an emergency department or an outpatient clinic. These experiences include patients with transplanted kidney, diabetes, prenatal care, emergency ophthalmological disorders, couple and family therapies, colorectal surgery, cancer, among others [8,9,10,11,12,13,14]. These practices emphasize the opportunities that telemedicine offers to maintain an uninterrupted follow-up care for complex patients, today and beyond the pandemic. Yet, some barriers have been identified: telemedicine does not fully replace face-to-face interactions, and increased privacy, regulatory and insurance coverage concerns must be addressed by policymakers. Additionally, more research is needed to assess its efficacy and quality of care it delivers [15,16].

The health threat caused by this virus also has particular implications for the vulnerable population—i.e., people living with disabilities, migrants, homeless, etc. [17]. These groups of people probably already live under disadvantageous conditions which have been aggravated by the pandemic, and they do not have access to telemedicine. Vulnerable-based proactive strategies need to be developed to cope with their specific needs. Additionally, the pandemic has brought serious mental health effects, worsening psychological distress at all ages [18,19]. This especially the case now, as there has been a significant impact on local economies given the strict measures imposed to contain the spread of the virus. This has resulted in isolation and increased unemployment rates and also affected insurance coverages [20].

Local governments will need to use appropriate data and consider their population characteristics and needs to help combat this virus. Therefore, it is imperative at this point to have a global view of the studies carried out since the COVID-19 pandemic started, to assess them and learn how to reduce the associated risks and improve access to healthcare services. The aim of this study is to map the existing literature of healthcare access after the COVID-19 pandemic using an ontological framework [21] to visualize the barriers to and drivers of access to healthcare and how to continue fighting the pandemic and having a health system accessible to all. Ontologies can describe relationships to model high-quality, linked and coherent data to share common understanding among people, and are a good holistic representation to simplify the available literature in the domain. As with any method, ontologies can have disadvantages. The structured natural language of the ontology may be unsuited to some researchers and contexts. It may not capture the full semantic range of a natural language narrative. Some of the weak signals in the natural language narrative may be lost in the process of structuring it. However, it is effective in providing a systemic view of a domain and addressing the issues systematically. The results from the ontology were also compared to the research coverage with the World Health Organization (WHO) guidelines on “Maintaining essential health services: operational guidance for the COVID-19 context” [6] to determine the overlapping and nonoverlapping areas. This analysis will help improve the feedback and learning from the translation of research to practice and of practice to research.

## 2. Materials and Methods

### 2.1. Ontology of Access to Healthcare during COVID-19

The ontology of access to healthcare during COVID-19 defines its dimensions, elements, and boundaries [22]. It deconstructs the policy problem’s complexity hierarchically [23], visualizes it in structured natural English, and encapsulates its combinatorial logic [24]. It organizes the terminologies, taxonomies, and narratives of the policy problem systemically, systematically, and symmetrically [25,26,27,28]. It is a cognitive map of the system [28,29,30,31] to: (a) design the policy alternatives, (b) determine effective, ineffective, and innovative policies, and (c) direct the choice through feedback and learning [32,33]. It is a qualitative theory [34] of the policy problem that can be used to describe the problem, explain its dynamics, predict the outcomes, and control the system through feedback and learning.

Similar ontologies have been used to conceptualize and analyze learning surveillance systems [35], mobile health (mHealth) [26], healthcare systems [36] and higher education policies [37]. The development and application of the ontology follows the description of the logic and process by Ramaprasad and Syn [22]. For this study, we borrowed and applied the ontology from a previously developed ontological framework of barriers to and facilitators of access to healthcare [21].

The ontology of access to healthcare during COVID-19 is shown in Figure 1. It encapsulates the various resources that affect access to healthcare such as spatial, temporal, financial, informational, human, and technological ones. These resources can be barriers, inhibitors, catalysts, or drivers to physical and virtual access to different types of healthcare. These forces could affect preventive care, wellness, episodic illness, chronic illness, rehabilitative, and palliative healthcare for different population segments such as the urban, rural, underprivileged, indigenous, disabled, and the elderly populations. Access to healthcare may be provided by varied personnel that include general physicians, specialist physicians, traditional healers, health workers, pharmacists, social workers, care providers, peers, and family.

### 2.2. Method

We visually synthesized the state of research in healthcare access during COVID-19 pandemic by mapping the research onto the ontology. The mapping was then used to generate the monad map and theme map to visualize the landscape of the domain. The visualization highlights the barriers to and drivers of access to healthcare during COVID-19.

The corpus of research was created from searching Scopus on TITLE-ABSTRACT-KEYWORDS of the articles indexed in the database. We experimented with different search terms. The broad term (healthcare w/3 access AND COVID-19) yielded 800 documents. The narrower term (health w/2 care w/3 access AND COVID-19) yielded 704 documents. Finally, the search term (healthcare AND access AND COVID-19) was used to retrieve 334 items on 9 September 2020. The items included different document types such as review, note, letter, conference paper, editorial, and other types of documents. We retained only 318 journal articles which represent a high-quality collection of peer-reviewed research on healthcare access during COVID-19. We further filtered out the selected articles with the word “access” in them. Based on the first iteration, the author with domain expertise further filtered 25 articles that were not relevant including protocols for hospital implementation. After this, all the authors agreed and further excluded 27 articles that were not related to healthcare access during COVID-19. Thus, 131 articles were included and considered for coding. Figure 2 details the search process and results, following the PRISMA reporting guidelines [38]. We then downloaded the title, abstract, and keywords of selected articles and imported them into an Excel spreadsheet for mapping. The reference management software Zotero (Corporation for Digital Scholarship, Vienna, VA, USA) was used to store the selected corpus.

The corpus of 131 articles was coded into the ontology through an iterative process between the three authors. The coding of all the articles went through two iterations by each of the three authors to ensure its reliability and validity. Further, we also used a glossary of elements to ensure the validity of coding. After the rounds of individual coding, the coders discussed the discrepancies in their coding and arrived at a consensus for the final coding. Only the dimensions and elements explicitly articulated in the title, abstract, and keywords were coded. Elements that were implicit in the section were not coded. The coding was binary (1 for present, 0 for absent) and was not scaled or weighted. In the analysis, both presence and absence of elements convey equally important information.

## 3. Results

The results of mapping the corpus onto the ontology are presented through a monad map (Figure 3) and a theme map (Figure 4). They are described next.

### 3.1. Monad Map

The monad map in Figure 3 numerically and visually summarizes the frequency of occurrence of each dimension and element of the ontology. The number adjacent to the dimension name and the element is the rate of occurrence in the 131 papers of access to healthcare during COVID-19 that were reviewed and mapped. The bar below each element is proportional to the frequency relative to the maximum frequency among all elements. Since each item can be coded to multiple elements of a dimension, the sum of the frequency of occurrence of elements may exceed the frequency of occurrence of the dimension to which the elements belong.

The dominant focus of the research was on the resources (126), force (124), and healthcare (117) during COVID-19. There is substantial focus on the type of access (108) and the personnel (70). There is less focus on the population type (46).

The research covers a spectrum of resources for access to healthcare and is heavily focused on temporal availability (77) and technological IT (52) resources. There is medium emphasis on informational educational (21), technological medical (20), and temporal scheduling (19). There is some emphasis on spatial distance (17), spatial location (16), financial expenditure (15), human psychology (15), human sociology (13), and financial income (12). The least emphasized resources are informational stimulant (8), technological transportation (6), and human cultural (3).

A significant proportion of articles consider the forces that affect access to healthcare. The most focus is on the barriers (72) to access; there is lesser emphasis on the catalysts (33) and drivers (32). There is little emphasis on inhibitors (17) to access. Specific forces, particularly barriers, received significant attention in the research. There is more attention on barriers than on drivers.

Although all the 131 articles are linked to healthcare, only 117 specify the type of care. The dominant focus is illness care-chronic (65) and -episodic (47). The next significant emphasis is on wellness (29) and preventive (27) care. Palliative (9) and rehabilitative (4) care are given little attention. Specific types of care have been given some attention in the research, whereas there has been relatively less paid to rising healthcare needs such as palliative care and rehabilitative care.

Again, although all the 131 articles are linked to access, only 108 specified the type of access. Physical access (76) has been emphasized the most and a few deals with virtual access (59). Specific types of access have not been given enough attention in the research. The focus has largely been on the traditional concept of access than not on the emergent perception.

Among personnel, the majority focus has been on specialist physicians (57), followed by nurses (26), general physicians (22), and health workers (21). The rest—family (5), social workers (3), pharmacists (2), care providers (2), and peers (1)—received little attention. There is no mention of traditional healers in the research.

Research focuses the least on the target population dimension. Among the different segments of the population, it largely focuses on the underprivileged population (32). The other population segments such as elderly (8), rural (6), urban (4), disabled (4), and indigenous (2) populations have not been given much attention in the research.

### 3.2. Theme Map

The theme map visually summarizes the co-occurrence of elements of the ontology in the population of articles. Hierarchical cluster analysis was done using SPSS (Statistical Package for Social Sciences; IBM: Chicago, IL, USA) with simple matching coefficient (SMC) as the distance measure and the nearest-neighbor aggregation procedure. SMC considers both presence (coded “1”) and absence (coded “0”) elements equally. The detailed rationale for the choice of the clustering method and the presentation of the results are given in Syn and Ramaprasad [39] and La Paz et al. [40]. The five themes represent the five equidistant clusters in the dendrogram of the agglomeration [39]. The colors in Figure 4 highlight the elements of the five themes.

The primary theme (in red), is the temporal availability barrier to physical access to chronic illness healthcare. The secondary theme (in brown) is access to episodic illness healthcare by specialist physicians. The tertiary theme (in yellow) is the technological IT catalyst/driver of virtual access to preventive/wellness healthcare for the underprivileged. The quaternary theme (in blue), is the spatial (distance/location)/temporal (scheduling)/financial (expenditure)/informational (educational)/human (psychological)/technological (medical) inhibitor to healthcare by general physicians/nurses/health workers. The quinary theme (no color), is financial (income)/informational (stimulant)/human (sociological/cultural)/technological (transportation) access to rehabilitative and palliative healthcare by traditional healers/pharmacists/social workers/care providers/peers/family for urban/rural/indigenous/disabled/elderly population.

The themes are in order of decreasing dominance in the research—the primary theme is the most emphasized and the quinary theme denotes nonexistence. The focus of the research is skewed to just a few, forces, types of access, resources, types of healthcare, and population segments. None of the themes comprehensively covers all the dimensions of the ontology. For example, the primary theme excludes personnel, and the secondary omits resources, force, access, and population. Overall, the research corpus coverage is segmented and not systemic.

## 4. Discussion

The ontology-based analysis of 131 research journal publications on access to healthcare during COVID-19 shows the thematic selectivity and segmentation in the research. Research in the primary theme is personnel and population agnostic. The theme shows the research emphasis on the temporal availability barrier to/of physical access to chronic illness. Availability to access chronic care has deteriorated due to diversion of medical specialists as “call of duty” for urgent COVID-19 cases [41]. The pandemic has further affected those seeking care for chronic conditions in areas without well-established telemedicine [9]. Telemedicine helps provide routine care for patients with chronic diseases who are at increased risk of severe illness if exposed to the virus. COVID-19 has made facility-based care for chronic conditions a major challenge. Chronic conditions such as chronic obstructive pulmonary disease, diabetes, and hypertension have been the most impacted due to decline in access to care [42]. During this time, it becomes essential to at least monitor and manage patients with chronic conditions and prioritize outpatient visits based on disease severity [41].

There is a significant contrast between the research on the primary theme and the WHO’s guidelines. While the research emphasizes the temporal availability barrier to/of physical access to chronic illness, the WHO operational guidelines of “Maintaining Essential Health Services” [5] addresses measures beyond availability of care for chronic conditions. Going beyond provision of medicines, supplies, and support from front-line workers, it calls for action on functional mapping health facilities for chronic, acute, and long-term care including those in private (commercial and nonprofit), public, and military systems. It supports the research in redesigning management strategies around limited availability of care providers. While research on the primary theme remains population agnostic, the WHO guidelines specify chronic care for children and the elderly. The guidelines move beyond teleconsultation and promote actions such as activating dedicated helplines and examining other outreach mechanisms. Importance of educating the chronic care patients on accessing telehealth, online services, and self-managing the condition brought out critical elements missing in research. Additionally, now that we are moving to a new stage of immunization, some guidelines have been established to prioritize the population that receives the vaccine, which depends on the distribution principle for equitable access and fair allocation defined by each country and may result in those living with chronic conditions being considered in a second stage of inoculation [43].

The secondary cluster indicates the research emphasis of episodic illness healthcare by specialists. The theme indicates a siloed focus on healthcare and personnel with emphasis only on episodic illness and specialists. Episodic illness and seeking care require the specialists to address the issues promptly to prevent aggravation. Addressing episodic illness through specialists care during this pandemic requires revamping of protocols so that there is standardization of outpatient activities with remote triage, protections, diagnostic tests, and precautions that allow provision of care while minimizing risk for both surgeons and patients [44]. With the diversion of all personnel resources, maximizing the availability of specialists for treatment of episodic illness requires adapting alternative treatment strategies [45].

With the research emphasis being siloed, the WHO guidelines give additional direction to make it a more systemic by providing a systematic approach. For episodic illness care, the guidelines highlight the need for time-sensitive interventions. They indicate having primary venues to address episodic care with settings that are suited for high-volume care. Modification of treatment pathways for specialist services through remote digital platforms during initial assessments is highlighted. The directions from the WHO further lay importance on prioritizing access for acute management of complications by considering repurposed facilities that ensure 24 h acute care. Overall, the guidelines cover more elements from the resource, force, and access dimensions, making it more systemic and systematic than the research corpus.

Even so, the guidelines raise significant ethical concerns, among which is the allocation of scarce resources. The guidelines consider the prioritization of patients to allocate scarce resources in relation to potential complications and specialist demand. However, given the pandemic, there are many kinds of scarce resources—i.e., infrastructure, general and specialist physicians, clinical resources, among others. This prioritization has aroused great resentment and triggered a public debate about the right to access healthcare services [46]. This was a matter of concern before the COVID-19 pandemic but has become more evident today. Although, there is a right for everyone to receive care, it is not feasible to overlook medical conditions and biological characteristics that differentiate one patient from another, which today has more relevance given the scarcity of resources faced by countries. Moreover, when we add to this equation the additional demand of services from the long-suffering COVID-19 patients from persistent medical conditions beyond the acute illness. Ensuring health equity is a challenge, and this pandemic has exposed the gaps existing worldwide and stressed public healthcare systems [47].

The third theme emphasizes technological IT catalyst/drivers of virtual access to preventive/wellness healthcare for the underprivileged. It covers important practice insights but neglects a comprehensive focus on different population segments. With the technological penetration in healthcare, today IT plays a critical role in acting as a catalyst or a driver for providing virtual care. Virtual clinics today with their technological tools are readily available for access and are used to deliver care. Deploying these mechanisms has provided high satisfaction to the patients and clinics are adopting this model, especially in resource-limited settings [9]. IT and virtual access to healthcare have extended from prevention and wellness care to other healthcare requirements such as prenatal care and wellness, with tailored telehealth regimens for surveillance and/or counseling [10]. Medical practice has changed in unprecedented ways and there is increased use of telemedicine services in safety and mental health, reproductive life planning, and routine screening for breast cancer [48]. Today, different technological approaches such as participatory digital contact notification is in practice for countries with limited access to healthcare resources and advanced technology [49].

The WHO guidelines emphasize digital modalities for various purposes to maintain the essential health services. The guidelines are in line with the current research and practice of using telemedicine solutions as catalysts and drivers, such as clinical consultations conducted via video chat or text message, e-pharmacies, staffed helplines, and mobile clinics with remote connections. They also support the practice of using digital health technologies as a proactive measure to manage their own health. The guidelines reflect the importance of prevention and wellness in terms of mental healthcare for populations such as school children and adolescents. They go beyond telecounselling and lay importance on follow-up with school dropouts and institute support mechanisms. The WHO guidelines broadly set the priority on wellness in terms of nutrition, monitoring status of noncommunicable diseases, and mental health. The research falls short in terms of the detailed approach taken towards and laid down by the WHO to maintain wellness of vulnerable populations.

The fourth/quaternary theme is siloed and segmented with the dominant focus on two dimensions—resources and personnel. There is also selective emphasis on the force. The theme is not comprehensive as it does not cover elements of type of access, healthcare, and population. The pandemic, in general, has brought into focus the utilization of available resources by the personnel in the health system. Research under this theme mainly focuses on the change in healthcare modalities that have been shifted to different forms and strategies by healthcare professionals [50]. Further, there is significant emphasis on the shortages in medical equipment and transfer of all human resources in addressing the pandemic which has led to revamping and redirection of resources through different triage approaches and prioritization [51].

The research emphasis in the quaternary theme aligns with the WHO guidelines in terms of the different resources employed to support timely action by the healthcare professionals. It highlights the repurposing of human, financial, and material resources, and mobilizing additional resources. Additionally, it aligns with the research emphasis on expenditure through reprogramming of budgetary resources, while monitoring expenditures to guarantee the effective use of resources and accountability [52].

The quinary cluster highlights many parts of pathways to healthcare access that have been missed in the research. There is limited research on crucial aspects such as sociological, cultural, and income resources that play significant role in accessing healthcare during the pandemic. Structural factors and societal factors concerning income, employment, health inequality, and racial bias add to the crisis [4]. Such factors call for a more comprehensive approach for access to care during COVID-19, with early testing, sustained, and affordable access to healthcare [53]. Further, the resources affecting healthcare access for different population segments is significantly neglected in research. Several social, environmental, and health risk factors have affected indigenous populations during this pandemic and strengthening of the health system with a community-based approach is vital [54]. Among the population segments, the elderly and disabled during this time of the pandemic are likely to require palliative care. Unique methods of health service delivery are necessary to ensure that vulnerable populations in underserviced metropolitan areas receive adequate and prompt palliative and rehabilitative care [55].

Some of the above research gaps are also amiss in the WHO guidelines. The cultural resources that would play a role in accessing healthcare during this time are not mentioned. Additionally, the roles of personnel such as traditional healers and social workers in maintaining the essential health services has been missed. While there are some elements that are not in focus in both research and guidelines, the guidelines address more elements present in the quinary theme.

Focusing on addressing the needs of marginalized populations, such as migrants and refugees, indigenous peoples, sex workers, and the homeless is given importance by the WHO. The guidelines lay detailed emphasis on maintaining essential health services and access to care for older people. It ranges from care of their mental health to rehabilitative and palliative care. In providing different types of care to the elderly the guidelines places importance on the role played by care givers, peers, and family. The pandemic has affected the mental health of all population segments including the personnel involved in healthcare [56,57]. The WHO guidelines on care for mental health are extensive. They go one step beyond and integrate psychological and sociological factors into providing psychosocial support for different population segments such as addicts, the elderly and school children.

The WHO guidelines are recommendations to ensure continuity in the access of essential care; however, each country may adapt them to their reality. Ontologies are this underexploited element of effective knowledge organization [58] that can help in their decision-making process. They can be used to:Provide a systemic view of the problem for advancing research and developing guidelines.Systematically analyze the emphases and gaps in research and practice and develop a balanced roadmap for both.Systematically analyze the gaps between research and practice and develop a strategy for effective translation between the two through feedback and learning.

## 5. Conclusions

For effective access to healthcare during pandemics such as COVID-19, the research and the guidelines must be systematically directed by a systemic framework. Further, the research must complement the guidelines and the guidelines must complement the research. Significant improvements can be made in the roadmaps for research and guidelines, as shown in the above analysis. The gaps in the research and the potential inclusions of practice guidelines gives the picture of the currently selective and segmented approaches in providing access to healthcare during COVID-19. While there are pathways unique to the research and in the practice guidelines, there is an overlap as well. A systemic ontology such as the one presented in this paper can promote a systematic approach to address the problems of access to healthcare during COVID-19 and similar pandemics. A systematic method for driving the research and guidelines will provide feedback on and help us learn about the gaps in the research, guidelines, and between the two. The feedback and learning will reduce the gaps and make both research and guidelines more effective. This study needs to be updated based on the new knowledge that is generated day by day with the development of the pandemic. However, it is a starting point for making informed decisions in public policy.

## Figures and Tables

**Figure 1 ijerph-18-02980-f001:**
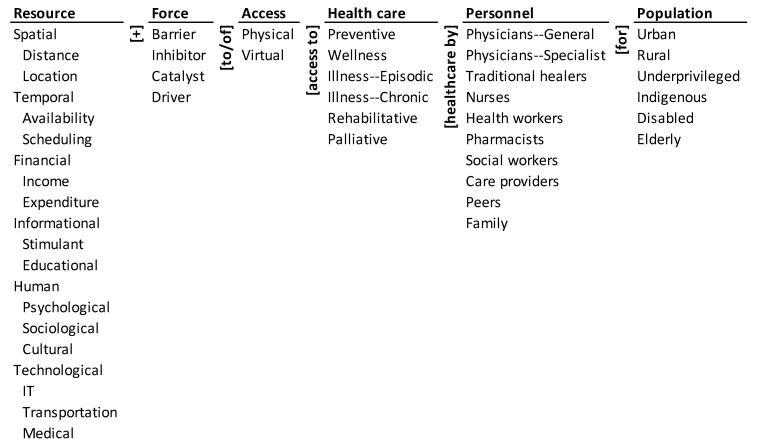
Ontology of access to healthcare during COVID-19. IT: Information Technology.

**Figure 2 ijerph-18-02980-f002:**
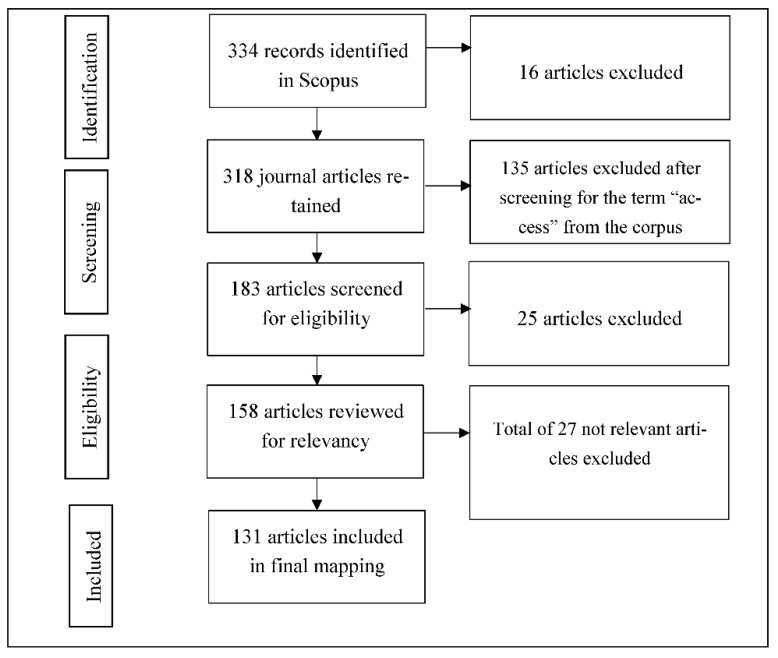
Search process and results.

**Figure 3 ijerph-18-02980-f003:**
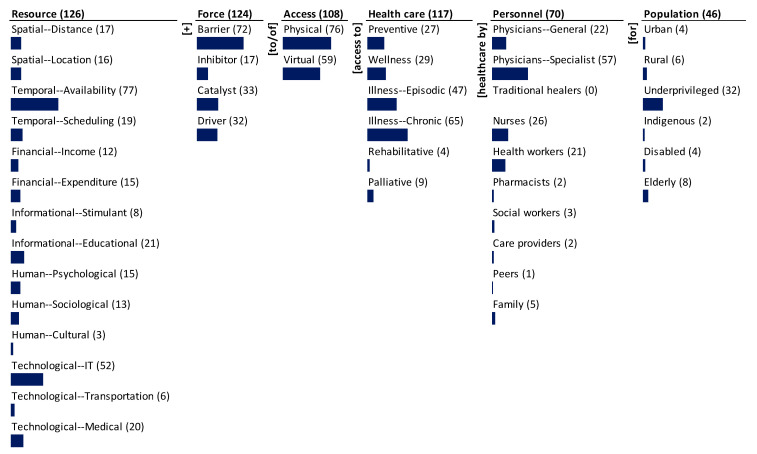
Monad map of research on access to healthcare during COVID-19.

**Figure 4 ijerph-18-02980-f004:**
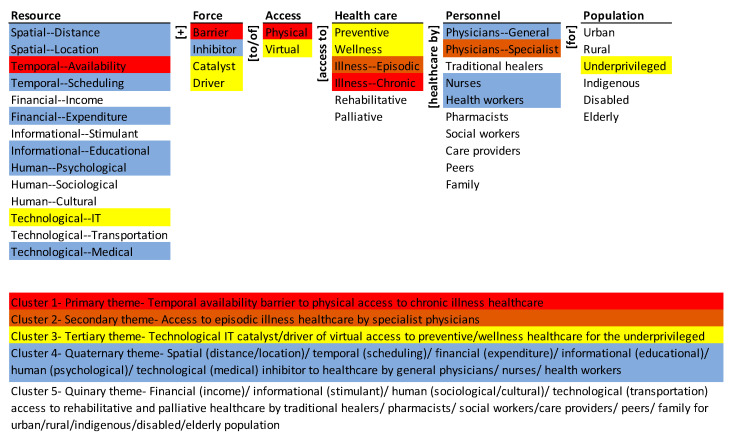
Theme map of the research on access to healthcare during COVID-19.

## Data Availability

The datasets used and/or analyzed during the current study are available from the corresponding author upon reasonable request.

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
