# Peer review of "Access to Healthcare during COVID-19"

_ijerph, 2021, doi:10.3390/ijerph18062980_

Round 1
Reviewer 1 Report
Dear Authors
I found the article “Access to Healthcare During COVID-19” very interesting, easy to read and showing a lot of hard work. Also, in general the references are adequate and updated. I congratulate the author for the well done job. Nevertheless, I have some minor comments, doubts and suggestion. I am sorry if I tend to be picky in some points.
In a more general overview I will call your attention to the following:
Page 1 - "Healthcare access refers to the level of health care services that the system can provide to the population." Can provide or the population can access? The offer can be very good but one cannot access the services due to mobility or socioeconomic issues, for instance. This sentence appears to contradict the following ones.
Page 2 - In my opinion we need a reference for "Also, the pandemic has brought serious mental health effects, worsening psychological distress at all ages."
Page 4 - Is not clear for me how the passage from 334 articles to 318 articles of high quality was done. It was a decision of one author or all? By majority or Unanimous? Authors chose the good ones and left 16 out or voted 16 as the bad ones. What parameters ruled this evaluation?
Page 4 - Authors state "into an Excel spreadsheet for mapping" but authors did not map anything. Obviously I am applying here some humorous. Just to call the attention for different backgrounds. Programmers use to call graphics (like those in excel) as maps, e.g. henon map, monad map and so on. For me, as a spatial modeler a map is something else.
Page 7 - "It is based on hierarchical cluster analysis of the elements, using the simple matching coefficient distance measure, and the nearest neighbor agglomeration". There are many different choices here that lead to very different results. I think these options need for a more detailed justification. For instance Hierarchical clustering has serious limitations and is somehow out of date. Nowadays we use advanced clustering methods such as Hybrid clustering, Fuzzy clustering, Model-based clustering or Density-based clustering (DBSCAN).
As for more narrow concerns I will emphasize:
Page 3 - Please replace WHO by World Health Organization (WHO)
Page 3 - References [21–25] and [26–28] in the text seem to be in Bold
Page 3 - Please replace mHealth by Mobile Health (mHealth)
Page 8 - Please replace IT by Information technology (IT). If it is IT I think is a little redundant to say it is technological
Page 8 - World Health Organization’s (WHO) should be defined previously. See earlier comment.
Best regards and continue the good work
Author Response
AUTHORS’ RESPONSES TO REVIEWER 1 COMMENTS
We thank the reviewer for your thoughtful suggestions to improve this paper. All of them reflect a willingness to engage with the basic argument and seek to improve it. We carefully considered the reviewer comments. Herein, we explain how we revised the manuscript based on your comments and recommendations.
The revision, based on the authors’ collective input, includes several positive changes. We hope that these revisions address or respond to the reviewer comments adequately, and have significantly improved the manuscript. Next, we offer detailed responses:
RESPONSES TO REVIEWER #1’s COMMENTS:
I found the article “Access to Healthcare During COVID-19” very interesting, easy to read and showing a lot of hard work. Also, in general the references are adequate and updated. I congratulate the author for the well done job. Nevertheless, I have some minor comments, doubts and suggestion. I am sorry if I tend to be picky in some points.
Answer: Thank you for all of your detailed comments and suggestions. We found them very useful as we approached our revision.
In a more general overview I will call your attention to the following:
Page 1 - "Healthcare access refers to the level of health care services that the system can provide to the population." Can provide or the population can access? The offer can be very good but one cannot access the services due to mobility or socioeconomic issues, for instance. This sentence appears to contradict the following ones.
Answer: We agreed with your comment, now the paper reads as follows:
“Healthcare access refers to the ease with which individuals can obtain needed health care. It is generally defined as the opportunity to use appropriate services in proportion to healthcare needs [1,2]. If services are available, then it exists an opportunity to obtain medical care, however, it is also limited by other barriers such as financial, organizational, social, cultural, etc. [3]. In this sense, the level of access influences the use of medical services, and therefore the health status of the population.”
Page 2 - In my opinion we need a reference for "Also, the pandemic has brought serious mental health effects, worsening psychological distress at all ages."
Answer: Agreed. We have cited two articles:
- Roy, A., Kumar Singh, A., Mishra, S., Chinnadurai, A., Mitra, A., Bakshi, O. Mental health implications of COVID-19 pandemic and its response in India. International Journal of Social Psychiatry 2020. https://doi.org/10.1177/0020764020950769.
- Javed, B., Sarwer, A., Soto, E. B., & Mashwani, Z. U. The coronavirus (COVID-19) pandemic's impact on mental health. The International journal of health planning and management 2020, 35(5), 993–996. https://doi.org/10.1002/hpm.3008.
Page 4 - Is not clear for me how the passage from 334 articles to 318 articles of high quality was done. It was a decision of one author or all? By majority or Unanimous? Authors chose the good ones and left 16 out or voted 16 as the bad ones. What parameters ruled this evaluation?
Answer: In the methodology, we have detailed out the process. 334 items included different document types such as review, note, letter, conference paper, editorial, and other types of documents. We retained only 318 journal articles which represent a high-quality collection of peer-reviewed research on healthcare access during COVID-19. We further filtered out the selected articles with the word “access” in them. Based on the first iteration the author with domain expertise further filtered 25 articles that were not relevant including protocols for hospital implementation. After this, all the authors agreed and further excluded articles that are not related to healthcare access during COVID-19. Thus, 131 articles were included and considered for coding. The PRISMA is updated accordingly. The paper reads as follows:
“Finally, the search term (healthcare AND access AND COVID-19) was used to retrieve 334 items on September 9, 2020. The items included different document types such as review, note, letter, conference paper, editorial, and other types of documents. We retained only 318 journal articles which represent a high-quality collection of peer-reviewed research on healthcare access during COVID-19. We further filtered out the selected articles with the word “access” in them. Based on the first iteration the author with domain expertise further filtered 25 articles that were not relevant including protocols for hospital implementation. After this, all the authors agreed and further excluded 27 articles that are not related to healthcare access during COVID-19. Thus, 131 articles were included and considered for coding. Figure 2 details the search process and results, following the PRISMA reporting guidelines [39].”
Page 4 - Authors state "into an Excel spreadsheet for mapping" but authors did not map anything. Obviously I am applying here some humorous. Just to call the attention for different backgrounds. Programmers use to call graphics (like those in excel) as maps, e.g. henon map, monad map and so on. For me, as a spatial modeler a map is something else.
Answer: We acknowledge the semantic differences between domains. We have been using the term in our previous publications using the method and have not been challenged. Perhaps we never had a spatial modeler as a reviewer. I hope the reviewer will pardon the semantic infringement on his/her domain.
Page 7 - "It is based on hierarchical cluster analysis of the elements, using the simple matching coefficient distance measure, and the nearest neighbor agglomeration". There are many different choices here that lead to very different results. I think these options need for a more detailed justification. For instance Hierarchical clustering has serious limitations and is somehow out of date. Nowadays we use advanced clustering methods such as Hybrid clustering, Fuzzy clustering, Model-based clustering or Density-based clustering (DBSCAN).
Answer: It has always been difficult for us to decide the detail of the method and the rationale for the choice. Some reviewers object strongly to too much ‘technical detail’, others to its absence. The extended response from two of our published papers is below. Instead of including the full rationale we shall refer to these two papers for the same:
“The clusters of ontological elements are derived in the hierarchical cluster analysis. The clusters are formed based on the coding similarity between pairs of ontological framework elements in the corpus measured by the simple matching coefficient (SMC) (Sokal and Michener, 1958). SMC is a symmetric similarity measure which considers presence (coded as “1”) and absence (coded as “0”) of elements in the articles equally, in contrast to other binary similarity/distance measures such as Jaccard (1912) and Sørensen Dice (1945) which only consider the presence of elements (Cheetham and Hazel, 1969; Gower, 1971). In our analysis, both presence and absence of elements in articles convey equally important information. SMC also provides a more consistent comparison across pairs of elements due to the fixed denominator. Hence, it is considered appropriate to measure the similarity of mappings between two articles for the cluster analysis which is used to descriptively summarize the data about the population of articles and not to make statistical inference about the population from a sample of articles. The clustering will be based on the single linkage algorithm. The agglomerative hierarchical clustering is preferred to K-means clustering and its variants because the primary purpose of the cluster analysis is to descriptively summarize the mappings of articles without any preconceptions about the clusters.”
Syn, T., & Ramaprasad, A. (2018). Megaprojects – Symbolic and Sublime: An Ontological Review. International Journal of Managing Projects in Business. https://doi.org/10.1108/IJMPB-03-2018-0054
“The dendrogram (Figure 4) is an exact visualization of these themes based on the association of elements in the data.” (p.13)
La Paz, A, Merigó, JM, Powell, P, Ramaprasad, A, Syn, T. Twenty‐five years of the Information Systems Journal: A bibliometric and ontological overview. Info Systems J. 2019; 1– 27. https://doi.org/10.1111/isj.12260
We have added the sentence: “The detailed rationale for the choice of the clustering method and the presentation of the results are given in Syn and Ramaprasad [39] and La Paz, et al. [40]”
As for more narrow concerns I will emphasize:
Page 3 - Please replace WHO by World Health Organization (WHO)
Answer: We have made the required changes.
Page 3 - References [21–25] and [26–28] in the text seem to be in Bold
Answer: We have changed it to normal text.
Page 3 - Please replace mHealth by Mobile Health (mHealth)
Answer: We now replaced it with mobile health (mHealth).
Page 8 - Please replace IT by Information technology (IT). If it is IT I think is a little redundant to say it is technological
Answer: We understand the redundancy, however, in the ontology we have defined under resources the Technological category with three subcategories: IT, Transportation and Medical. In this way, we refer to technological IT, technological Transportation, and Technological Medical. To make it clearer we have added a hyphen to separate the terms, now it reads Technological-IT.
Page 8 - World Health Organization’s (WHO) should be defined previously. See earlier comment.
Answer: Now, we defined WHO the first time that is mentioned in the text.

Reviewer 2 Report
Figure 2 representing the search process must respect the PRISMA guidelines: Identification, Screening, Eligibility and Included, with details about the reasons for exclusion.
Figure 4 - explain what are the meaning of the colors near the figure, not only in the text.
What are the strengths and the limits of your research?
Where did you include articles about the impact of the COVID-19 on training among young physicians, as the next one?
Ungureanu, B.S.; Vladut, C.; Bende, F.; Sandru, V.; Tocia, V.; Turcu-Stiolica, R.-V.; Groza, A.; Balan, G.G.; Turcu-Stiolica, A. Impact of the COVID-19 Pandemic on Health-Related Quality of Life, Anxiety, and Training Among Young Gastroenterologists in Romania. Psychol. 2020, 11, 579177. doi: 10.3389/fpsyg.2020.579177
Author Response
AUTHORS’ RESPONSES TO REVIEWER 2 COMMENTS
We thank the reviewer for your thoughtful suggestions to improve this paper. All of them reflect a willingness to engage with the basic argument and seek to improve it. We carefully considered the reviewer comments. Herein, we explain how we revised the manuscript based on your comments and recommendations.
The revision, based on the authors’ collective input, includes several positive changes. We hope that these revisions address or respond to the reviewer comments adequately, and have significantly improved the manuscript. Next, we offer detailed responses:
RESPONSES TO REVIEWER #2’s COMMENTS:
Figure 2 representing the search process must respect the PRISMA guidelines: Identification, Screening, Eligibility and Included, with details about the reasons for exclusion.
Answer: In this new version of the manuscript, we have detailed out the reasons for exclusion and updated the PRISMA. See figure 2.
Figure 4 - explain what are the meaning of the colors near the figure, not only in the text.
Answer: We have added the meaning of colors in figure 4.
What are the strengths and the limits of your research?
Answer: In the conclusion we have made explicit the strengths and limits of our research:
“A systemic ontology such as the one presented in this paper can promote a systematic approach to address the problems of access to healthcare during COVID-19 and similar pandemics. A systematic method for driving the research and guidelines will provide feedback on and help learn about the gaps in the research, guidelines, and between the two. The feedback and learning will reduce the gaps and make both research and guidelines more effective. This study needs to be updated based on the new knowledge that is generated day by day with the development of the pandemic. However, it is a starting point for making informed decisions in public policy.”
Where did you include articles about the impact of the COVID-19 on training among young physicians, as the next one?
Ungureanu, B.S.; Vladut, C.; Bende, F.; Sandru, V.; Tocia, V.; Turcu-Stiolica, R.-V.; Groza, A.; Balan, G.G.; Turcu-Stiolica, A. Impact of the COVID-19 Pandemic on Health-Related Quality of Life, Anxiety, and Training Among Young Gastroenterologists in Romania. Psychol. 2020, 11, 579177. doi: 10.3389/fpsyg.2020.579177
Answer: We have included a line in the discussion section and cited the reference mentioned. The manuscript now reads:
“The pandemic has affected the mental health of all population segments including the personnel involved in healthcare [56,57]”

Reviewer 3 Report
The major objective of this study is to map the existing literature of healthcare access after the COVID-19 pandemic using an ontological framework. Overall, the current version of this paper is very weak, so I recommend being rejected. The comments are listed as below:
- The authors extracted the healthcare access information from 131 articles. Firstly, when and where these articles are talked about. The authors downloaded the title, abstract, and keywords of selected articles to map. Did you review the whole articles? How to validate the accuracy of your results? For different countries, the healthcare access is totally different. The authors do not clarify.
- The Introduction is very weak. It mainly talks about the background of healthcare and COVID-19. As a scientific paper, you should review the methods such as how to map the healthcare access, the advantages, and disadvantages, why do you choose ontological framework…
- The method is also weak, and please see the first comment. Figure 2 is not essential.
- The results are very basic and not validated.
Author Response
AUTHORS’ RESPONSES TO REVIEWER 3 COMMENTS
We thank the reviewer for your thoughtful suggestions to improve this paper. All of them reflect a willingness to engage with the basic argument and seek to improve it. We carefully considered the reviewer comments. Herein, we explain how we revised the manuscript based on your comments and recommendations.
The revision, based on the authors’ collective input, includes several positive changes. We hope that these revisions address or respond to the reviewer comments adequately, and have significantly improved the manuscript. Next, we offer detailed responses:
RESPONSES TO REVIEWER #3’s COMMENTS:
The major objective of this study is to map the existing literature of healthcare access after the COVID-19 pandemic using an ontological framework. Overall, the current version of this paper is very weak, so I recommend being rejected. The comments are listed as below:
1. The authors extracted the healthcare access information from 131 articles. Firstly, when and where these articles are talked about. The authors downloaded the title, abstract, and keywords of selected articles to map. Did you review the whole articles? How to validate the accuracy of your results? For different countries, the healthcare access is totally different. The authors do not clarify.
Answer: We have strengthened the methodology section to answer to this comment. The manuscript includes articles that were retrieve until September 9, 2020. We downloaded title, abstract and keywords to an excel sheet that was used to code the ontology. A key step in the process of any systematic review is citation screening, which involves manual review of titles and abstracts to identify potentially eligible articles for inclusion in the review process. This screening is time-consuming yet a crucial aspect of the ontology process, since failure to identify relevant studies can affect the validity of our review. To increase the reliability of article selection, we use three reviewers, one more that what is often recommended.
We understand that healthcare access differs between countries, we mapped what has been researched about healthcare access and COVID 19 in the literature, which gives you a picture, a global perspective. Then, we use this research to compare it with WHO guidelines, those guidelines are also global guidelines, you need to adapt it depending on the country. The same with our study. Moreover, you can use this ontology and make a country-specific study about healthcare access. Now, the paper reads as follows:
“Finally, the search term (healthcare AND access AND COVID-19) was used to retrieve 334 items articles on September 9, 2020. The items included different document types such as review, note, letter, conference paper, editorial, and other types of documents. We retained only 318 journal articles which represent a high-quality collection of peer-reviewed research on healthcare access during COVID-19. We further filtered out the selected articles with the word “access” in them. Based on the first iteration the author with domain expertise further filtered 25 articles that were not relevant including protocols for hospital implementation. After this, all the authors agreed and further excluded 27 articles that are not related to healthcare access during COVID-19. Thus, 131 articles were included and considered for coding. Figure 2 details the search process and results, following the PRISMA reporting guidelines [39].”
2. The Introduction is very weak. It mainly talks about the background of healthcare and COVID-19. As a scientific paper, you should review the methods such as how to map the healthcare access, the advantages, and disadvantages, why do you choose ontological framework…
Answer: As mentioned in the previous answer, we reviewed the method section as recommended. This is a paper that uses an already developed ontology, which is explained in the following publication and cited in our manuscript:
Núñez, A., Ramaprasad, A., Syn, T., & Lopez, H. An Ontological Analysis of the Barriers to and Facilitators of Access to Health Care. Journal of Public Health 2020. https://doi.org/10.1007/s10389-020-01265-4.
We also add the reasons why we chose an ontology in the introduction, arguing that:
“Ontologies can describe relationships for modeling high-quality, linked and coherent data to share common understanding among people, are a good holistic representation to simplify the available literature on the domain. As with any method, ontologies can have disadvantages. The structured natural language of the ontology may be unsuited to some researchers and contexts. It may not capture the full semantic range of a natural language narrative. Some of the weak signals in the natural language narrative may be lost in the process of structuring it. However, it is effective in providing a systemic view of a domain and addressing the issues systematically.”
3. The method is also weak, and please see the first comment. Figure 2 is not essential.
Answer: As mentioned before, the method section has been explained better based on the comments from all the reviewers. Figure 2 details the search process and results, following the PRISMA reporting guidelines, which has been modified to clarify the methodology.
4. The results are very basic and not validated.
Answer: In systematic review, one way to validate the results is using two reviewers, we included three reviewers to make the results more valid and reliable. Further, we used a glossary of elements to assure the validity of coding. Moreover, we use the WHO guidelines to check the coherence of our results and the policies suggested for implementation in different countries. The results are useful in public policy especially because guidelines not always consider the up-to-date research available. Many times, we lack integration between academic research and the development of policies. This ontology is very useful to bring these two areas together and make this integration happen.

Round 2
Reviewer 3 Report
I do not have further questions after the revisions. Thank you!